# Stochastic Optimization for
# Large-scale Optimal Transport

**Aude Genevay**
CEREMADE, Université Paris-Dauphine
INRIA – Mokaplan project-team
genevay@ceremade.dauphine.fr

**Marco Cuturi**
CREST, ENSAE
Université Paris-Saclay
marco.cuturi@ensae.fr

**Gabriel Peyré**
CNRS and DMA, École Normale Supérieure
INRIA – Mokaplan project-team
gabriel.peyre@ens.fr

**Francis Bach**
INRIA – Sierra project-team
DI, ENS
francis.bach@inria.fr

## Abstract

Optimal transport (OT) defines a powerful framework to compare probability distributions in a geometrically faithful way. However, the practical impact of OT is still limited because of its computational burden. We propose a new class of stochastic optimization algorithms to cope with large-scale OT problems. These methods can handle arbitrary distributions (either discrete or continuous) as long as one is able to draw samples from them, which is the typical setup in high-dimensional learning problems. This alleviates the need to discretize these densities, while giving access to provably convergent methods that output the correct distance without discretization error. These algorithms rely on two main ideas: *(a)* the dual OT problem can be re-cast as the maximization of an expectation; *(b)* the entropic regularization of the primal OT problem yields a smooth dual optimization which can be addressed with algorithms that have a provably faster convergence. We instantiate these ideas in three different setups: *(i)* when comparing a discrete distribution to another, we show that incremental stochastic optimization schemes can beat Sinkhorn's algorithm, the current state-of-the-art finite dimensional OT solver; *(ii)* when comparing a discrete distribution to a continuous density, a semi-discrete reformulation of the dual program is amenable to averaged stochastic gradient descent, leading to better performance than approximately solving the problem by discretization ; *(iii)* when dealing with two continuous densities, we propose a stochastic gradient descent over a reproducing kernel Hilbert space (RKHS). This is currently the only known method to solve this problem, apart from computing OT on finite samples. We backup these claims on a set of discrete, semi-discrete and continuous benchmark problems.

## 1 Introduction

Many problems in computational sciences require to compare probability measures or histograms. As a set of representative examples, let us quote: bag-of-visual-words comparison in computer vision [17], color and shape processing in computer graphics [21], bag-of-words for natural language processing [11] and multi-label classification [9]. In all of these problems, a geometry between the features (words, visual words, labels) is usually known, and can be leveraged to compare probability distributions in a geometrically faithful way. This underlying geometry might be for instance the planar Euclidean domain for 2-D shapes, a perceptual 3D color metric space for image processing or a high-dimensional semantic embedding for words. Optimal transport (OT) [24] is the canonical

way to automatically lift this geometry to define a metric for probability distributions. That metric is known as the *Wasserstein* or *earth mover's* distance. As an illustrative example, OT can use a metric between words to build a metric between documents that are represented as frequency histograms of words (see [11] for details). All the above-cited lines of work advocate, among others, that OT is the natural choice to solve these problems, and that it leads to performance improvement when compared to geometrically-oblivious distances such as the Euclidean or $\chi^2$ distances or the Kullback-Leibler divergence. However, these advantages come at the price of an enormous computational overhead. This is especially true because current OT solvers require to sample beforehand these distributions on a pre-defined set of points, or on a grid. This is both inefficient (in term of storage and speed) and counter-intuitive. Indeed, most high-dimensional computational scenarios naturally represent distributions as objects from which one can *sample*, not as density functions to be discretized. Our goal is to alleviate these shortcomings. We propose a class of provably convergent stochastic optimization schemes that can handle both discrete and continuous distributions through sampling.

**Previous works.** The prevalent way to compute OT distances is by solving the so-called Kantorovitch problem [10] (see Section 2 for a short primer on the basics of OT formulations), which boils down to a large-scale linear program when dealing with discrete distributions (i.e., finite weighted sums of Dirac masses). This linear program can be solved using network flow solvers, which can be further refined to assignment problems when comparing measures of the same size with uniform weights [3]. Recently, regularized approaches that solve the OT with an entropic penalization [6] have been shown to be extremely efficient to approximate OT solutions at a very low computational cost. These regularized approaches have supported recent applications of OT to computer graphics [21] and machine learning [9]. These methods apply the celebrated Sinkhorn algorithm [20], and can be extended to solve more exotic transportation-related problems such as the computation of barycenters [21]. Their chief computational advantage over competing solvers is that each iteration boils down to matrix-vector multiplications, which can be easily parallelized, streams extremely well on GPU, and enjoys linear-time implementation on regular grids or triangulated domains [21].

These methods are however purely discrete and cannot cope with continuous densities. The only known class of methods that can overcome this limitation are so-called semi-discrete solvers [1], that can be implemented efficiently using computational geometry primitives [12]. They can compute distance between a discrete distribution and a continuous density. Nonetheless, they are restricted to the Euclidean squared cost, and can only be implemented in low dimensions (2-D and 3-D). Solving these semi-discrete problems efficiently could have a significant impact for applications to density fitting with an OT loss [2] for machine learning applications, see [13]. Lastly, let us point out that there is currently no method that can compute OT distances between two continuous densities, which is thus an open problem we tackle in this article.

**Contributions.** This paper introduces stochastic optimization methods to compute large-scale optimal transport in all three possible settings: *discrete* OT, to compare a discrete *vs.* another discrete measure; *semi-discrete* OT, to compare a discrete *vs.* a continuous measure; and *continous* OT, to compare a continuous *vs.* another continuous measure. These methods can be used to solve classical OT problems, but they enjoy faster convergence properties when considering their entropic-regularized versions. We show that the discrete regularized OT problem can be tackled using incremental algorithms, and we consider in particular the stochastic averaged gradient (SAG) method [19]. Each iteration of that algorithm requires $N$ operations ($N$ being the size of the supports of the input distributions), which makes it scale better in large-scale problems than the state-of-the-art Sinkhorn algorithm, while still enjoying a convergence rate of $O(1/k)$, $k$ being the number of iterations. We show that the semi-discrete OT problem can be solved using averaged stochastic gradient descent (SGD), whose convergence rate is $O(1/\sqrt{k})$. This approach is numerically advantageous over the brute force approach consisting in sampling first the continuous density to solve next a discrete OT problem. Lastly, for continuous optimal transport, we propose a novel method which makes use of an expansion of the dual variables in a reproducing kernel Hilbert space (RKHS). This allows us for the first time to compute with a converging algorithm OT distances between two arbitrary densities, under the assumption that the two potentials belong to such an RKHS.

**Notations.** In the following we consider two metric spaces $\mathcal{X}$ and $\mathcal{Y}$. We denote by $\mathcal{M}_+^1(\mathcal{X})$ the set of positive Radon probability measures on $\mathcal{X}$, and $\mathcal{C}(\mathcal{X})$ the space of continuous functions on $\mathcal{X}$. Let $\mu \in \mathcal{M}_+^1(\mathcal{X})$, $\nu \in \mathcal{M}_+^1(\mathcal{Y})$, we define

$$\Pi(\mu, \nu) \overset{\text{def.}}{=} \left\{ \pi \in \mathcal{M}_+^1(\mathcal{X} \times \mathcal{Y}) \, ; \, \forall (A, B) \subset \mathcal{X} \times \mathcal{Y}, \pi(A \times \mathcal{Y}) = \mu(A), \pi(\mathcal{X} \times B) = \nu(B) \right\},$$

the set of joint probability measures on $\mathcal{X} \times \mathcal{Y}$ with marginals $\mu$ and $\nu$. The Kullback-Leibler divergence between joint probabilities is defined as

$$\forall (\pi, \xi) \in \mathcal{M}_+^1(\mathcal{X} \times \mathcal{Y})^2, \quad \mathrm{KL}(\pi|\xi) \stackrel{\text{def.}}{=} \int_{\mathcal{X} \times \mathcal{Y}} \left( \log\left( \tfrac{\mathrm{d}\pi}{\mathrm{d}\xi}(x, y) \right) - 1 \right) \mathrm{d}\xi(x, y),$$

where we denote $\frac{\mathrm{d}\pi}{\mathrm{d}\xi}$ the relative density of $\pi$ with respect to $\xi$, and by convention $\mathrm{KL}(\pi|\xi) \stackrel{\text{def.}}{=} +\infty$ if $\pi$ does not have a density with respect to $\xi$. The Dirac measure at point $x$ is $\delta_x$. For a set $C$, $\iota_C(x) = 0$ if $x \in C$ and $\iota_C(x) = +\infty$ otherwise. The probability simplex of $N$ bins is $\Sigma_N = \left\{ \boldsymbol{\mu} \in \mathbb{R}_+^N \; ; \; \sum_i \boldsymbol{\mu}_i = 1 \right\}$. Element-wise multiplication of vectors is denoted by $\odot$ and $K^\top$ denotes the transpose of a matrix $K$. We denote $\mathbb{1}_N = (1, \ldots, 1)^\top \in \mathbb{R}^N$ and $\mathbb{0}_N = (0, \ldots, 0)^\top \in \mathbb{R}^N$.

## 2 Optimal Transport: Primal, Dual and Semi-dual Formulations

We consider the optimal transport problem between two measures $\mu \in \mathcal{M}_+^1(\mathcal{X})$ and $\nu \in \mathcal{M}_+^1(\mathcal{Y})$, defined on metric spaces $\mathcal{X}$ and $\mathcal{Y}$. No particular assumption is made on the form of $\mu$ and $\nu$, we only assume that they both can be sampled from to be able to apply our algorithms.

**Primal, Dual and Semi-dual Formulations.** The Kantorovich formulation [10] of OT and its entropic regularization [6] can be conveniently written in a single convex optimization problem as follows

$$\forall (\mu, \nu) \in \mathcal{M}_+^1(\mathcal{X}) \times \mathcal{M}_+^1(\mathcal{Y}), \ W_\varepsilon(\mu, \nu) \stackrel{\text{def.}}{=} \min_{\pi \in \Pi(\mu, \nu)} \int_{\mathcal{X} \times \mathcal{Y}} c(x, y) \mathrm{d}\pi(x, y) + \varepsilon \, \mathrm{KL}(\pi|\mu \otimes \nu). \quad (\mathcal{P}_\varepsilon)$$

Here $c \in \mathcal{C}(\mathcal{X} \times \mathcal{Y})$ and $c(x, y)$ should be interpreted as the "ground cost" to move a unit of mass from $x$ to $y$. This $c$ is typically application-dependent, and reflects some prior knowledge on the data to process. We refer to the introduction for a list of previous work where various examples (in imaging, vision, graphics or machine learning) of such costs are given.

When $\mathcal{X} = \mathcal{Y}$, $\varepsilon = 0$ and $c = d^p$ for $p \geq 1$, where $d$ is a distance on $\mathcal{X}$, then $W_0(\mu, \nu)^{\frac{1}{p}}$ is known as the $p$-Wasserstein distance on $\mathcal{M}_+^1(\mathcal{X})$. Note that this definition can be used for any type of measure, both discrete and continuous. When $\varepsilon > 0$, problem $(\mathcal{P}_\varepsilon)$ is strongly convex, so that the optimal $\pi$ is unique, and algebraic properties of the KL regularization result in computations that can be tackled using the Sinkhorn algorithm [6].

For any $c \in \mathcal{C}(\mathcal{X} \times \mathcal{Y})$, we define the following constraint set

$$U_c \stackrel{\text{def.}}{=} \left\{ (u, v) \in \mathcal{C}(\mathcal{X}) \times \mathcal{C}(\mathcal{Y}) \; ; \; \forall (x, y) \in \mathcal{X} \times \mathcal{Y}, u(x) + v(y) \leq c(x, y) \right\},$$

and define its indicator function as well as its "smoothed" approximation

$$\iota_{U_c}^\varepsilon(u, v) \stackrel{\text{def.}}{=} \begin{cases} \iota_{U_c}(u, v) & \text{if} \quad \varepsilon = 0, \\ \varepsilon \int_{\mathcal{X} \times \mathcal{Y}} \exp(\tfrac{u(x) + v(y) - c(x, y)}{\varepsilon}) \mathrm{d}\mu(x) \mathrm{d}\nu(y) & \text{if} \quad \varepsilon > 0. \end{cases} \quad (1)$$

For any $v \in \mathcal{C}(\mathcal{Y})$, we define its $c$-transform and its "smoothed" approximation

$$\forall x \in \mathcal{X}, \quad v^{c, \varepsilon}(x) \stackrel{\text{def.}}{=} \begin{cases} \min_{y \in \mathcal{Y}} c(x, y) - v(y) & \text{if} \quad \varepsilon = 0, \\ -\varepsilon \log \left( \int_{\mathcal{Y}} \exp(\tfrac{v(y) - c(x, y)}{\varepsilon}) \mathrm{d}\nu(y) \right) & \text{if} \quad \varepsilon > 0. \end{cases} \quad (2)$$

The proposition below describes two dual problems. It is central to our analysis and paves the way for the application of stochastic optimization methods.

**Proposition 2.1** (Dual and semi-dual formulations). *For $\varepsilon \geq 0$, one has*

$$W_\varepsilon(\mu, \nu) = \max_{u \in \mathcal{C}(\mathcal{X}), v \in \mathcal{C}(\mathcal{Y})} F_\varepsilon(u, v) \stackrel{\text{def.}}{=} \int_{\mathcal{X}} u(x) \mathrm{d}\mu(x) + \int_{\mathcal{Y}} v(y) \mathrm{d}\nu(y) - \iota_{U_c}^\varepsilon(u, v), \quad (\mathcal{D}_\varepsilon)$$

$$= \max_{v \in \mathcal{C}(\mathcal{Y})} H_\varepsilon(v) \stackrel{\text{def.}}{=} \int_{\mathcal{X}} v^{c, \varepsilon}(x) \mathrm{d}\mu(x) + \int_{\mathcal{Y}} v(y) \mathrm{d}\nu(y) - \varepsilon, \quad (\mathcal{S}_\varepsilon)$$

*where $\iota_{U_c}^\varepsilon$ is defined in (1) and $v^{c, \varepsilon}$ in (2). Furthermore, $u$ solving $(\mathcal{D}_\varepsilon)$ is recovered from an optimal $v$ solving $(\mathcal{S}_\varepsilon)$ as $u = v^{c, \varepsilon}$. For $\varepsilon > 0$, the solution $\pi$ of $(\mathcal{P}_\varepsilon)$ is recovered from any $(u, v)$ solving $(\mathcal{D}_\varepsilon)$ as $\mathrm{d}\pi(x, y) = \exp(\tfrac{u(x) + v(y) - c(x, y)}{\varepsilon}) \mathrm{d}\mu(x) \mathrm{d}\nu(y)$.*

*Proof.* Problem $(\mathcal{D}_\varepsilon)$ is the convex dual of $(\mathcal{P}_\varepsilon)$, and is derived using Fenchel-Rockafellar's theorem. The relation between $u$ and $v$ is obtained by writing the first order optimality condition for $v$ in $(\mathcal{D}_\varepsilon)$. Plugging this expression back in $(\mathcal{D}_\varepsilon)$ yields $(\mathcal{S}_\varepsilon)$. $\qquad\qquad\square$

Problem $(\mathcal{P}_\varepsilon)$ is called the primal while $(\mathcal{D}_\varepsilon)$ is its associated dual problem. We refer to $(\mathcal{S}_\varepsilon)$ as the "semi-dual" problem, because in the special case $\varepsilon = 0$, $(\mathcal{S}_\varepsilon)$ boils down to the so-called semi-discrete OT problem [1]. Both dual problems are concave maximization problems. The optimal dual variables $(u, v)$—known as Kantorovitch potentials—are not unique, since for any solution $(u, v)$ of $(\mathcal{D}_\varepsilon)$, $(u + \lambda, v - \lambda)$ is also a solution for any $\lambda \in \mathbb{R}$. When $\varepsilon > 0$, they can be shown to be unique up to this scalar translation [6]. We refer to the supplementary material for a discussion (and proofs) of the convergence of the solutions of $(\mathcal{P}_\varepsilon)$, $(\mathcal{D}_\varepsilon)$ and $(\mathcal{S}_\varepsilon)$ towards those of $(\mathcal{P}_0)$, $(\mathcal{D}_0)$ and $(\mathcal{S}_0)$ as $\varepsilon \to 0$.

A key advantage of $(\mathcal{S}_\varepsilon)$ over $(\mathcal{D}_\varepsilon)$ is that, when $\nu$ is a discrete density (but not necessarily $\mu$), then $(\mathcal{S}_\varepsilon)$ is a finite-dimensional concave maximization problem, which can thus be solved using stochastic programming techniques, as highlighted in Section 4. By contrast, when both $\mu$ and $\nu$ are continuous densities, these dual problems are intrinsically infinite dimensional, and we propose in Section 5 more advanced techniques based on RKHSs.

**Stochastic Optimization Formulations.** The fundamental property needed to apply stochastic programming is that both dual problems $(\mathcal{D}_\varepsilon)$ and $(\mathcal{S}_\varepsilon)$ must be rephrased as maximizing expectations:

$$\forall \varepsilon > 0, \ F_\varepsilon(u, v) = \mathbb{E}_{X,Y}\left[f_\varepsilon(X, Y, u, v)\right] \quad \text{and} \quad \forall \varepsilon \geq 0, \ H_\varepsilon(v) = \mathbb{E}_X\left[h_\varepsilon(X, v)\right], \qquad (3)$$

where the random variables $X$ and $Y$ are independent and distributed according to $\mu$ and $\nu$ respectively, and where, for $(x, y) \in \mathcal{X} \times \mathcal{Y}$ and $(u, v) \in \mathcal{C}(\mathcal{X}) \times \mathcal{C}(\mathcal{Y})$,

$$\forall \varepsilon > 0, \quad f_\varepsilon(x, y, u, v) \stackrel{\text{def.}}{=} u(x) + v(y) - \varepsilon \exp\left(\frac{u(x) + v(y) - c(x, y)}{\varepsilon}\right),$$

$$\forall \varepsilon \geq 0, \quad h_\varepsilon(x, v) \stackrel{\text{def.}}{=} \int_{\mathcal{Y}} v(y)\mathrm{d}\nu(y) + v^{c,\varepsilon}(x) - \varepsilon.$$

This reformulation is at the heart of the methods detailed in the remainder of this article. Note that the dual problem $(\mathcal{D}_\varepsilon)$ cannot be cast as an unconstrained expectation maximization problem when $\varepsilon = 0$, because of the constraint on the potentials which arises in that case.

When $\nu$ is discrete, i.e $\nu = \sum_{j=1}^J \boldsymbol{\nu}_j \delta_{y_j}$ the potential $v$ is a $J$-dimensional vector $(\mathbf{v}_j)_{j=\{1\dots J\}}$ and we can compute the gradient of $h_\varepsilon$. When $\varepsilon > 0$ the gradient reads $\nabla_v h_\varepsilon(v, x) = \boldsymbol{\nu} - \pi(x)$ and the hessian is given by $\partial_v^2 h_\varepsilon(v, x) = \frac{1}{\varepsilon}(\pi(x)\pi(x)^T - \mathrm{diag}(\pi(x)))$ where $\pi(x)_i = \exp(\frac{\mathbf{v}_i - c(x, y_i)}{\varepsilon})\left(\sum_{j=1}^J \exp(\frac{\mathbf{v}_j - c(x, y_j)}{\varepsilon})\right)^{-1}$. The eigenvalues of the hessian can be upper bounded by $\frac{1}{\varepsilon}$, which guarantees a lipschitz gradient, and lower bounded by $0$ which does not ensure strong convexity. In the unregularized case, $h_0$ is not smooth and a subgradient is given by $\nabla_v h_0(v, x) = \boldsymbol{\nu} - \tilde{\pi}(x)$, where $\tilde{\pi}(x)_i = \chi_{i=j^\star}$ and $j^\star = \arg\min_{j\in\{1\dots J\}} c(x, y_j) - \mathbf{v}_j$ (when several elements are in the argmin, we arbitrarily choose one of them to be $j^\star$). We insist on the lack of strong convexity of the semi-dual problem, as it impacts the convergence properties of the stochastic algorithms (stochastic averaged gradient and stochastic gradient descent) described below.

## 3 Discrete Optimal Transport

We assume in this section that both $\mu$ and $\nu$ are discrete measures, i.e. finite sums of Diracs, of the form $\mu = \sum_{i=1}^I \boldsymbol{\mu}_i \delta_{x_i}$ and $\nu = \sum_{j=1}^J \boldsymbol{\nu}_j \delta_{y_j}$, where $(x_i)_i \subset \mathcal{X}$ and $(y_j)_j \subset \mathcal{Y}$, and the histogram vector weights are $\boldsymbol{\mu} \in \Sigma_I$ and $\boldsymbol{\nu} \in \Sigma_J$. These discrete measures may come from the evaluation of continuous densities on a grid, counting features in a structured object, or be empirical measures based on samples. This setting is relevant for several applications, including all known applications of the earth mover's distance. We show in this section that our stochastic formulation can prove extremely efficient to compare measures with a large number of points.

**Discrete Optimization and Sinkhorn.** In this setup, the primal $(\mathcal{P}_\varepsilon)$, dual $(\mathcal{D}_\varepsilon)$ and semi-dual $(\mathcal{S}_\varepsilon)$ problems can be rewritten as finite-dimensional optimization problems involving the cost matrix

$\mathbf{c} \in \mathbb{R}_+^{I \times J}$ defined by $\mathbf{c}_{i,j} = c(x_i, y_j)$:

$$W_\varepsilon(\mu, \nu) = \min_{\boldsymbol{\pi} \in \mathbb{R}_+^{I \times J}} \left\{ \sum_{i,j} \mathbf{c}_{i,j} \boldsymbol{\pi}_{i,j} + \varepsilon \sum_{i,j} \left( \log \frac{\boldsymbol{\pi}_{i,j}}{\boldsymbol{\mu}_i \boldsymbol{\nu}_j} - 1 \right) \boldsymbol{\pi}_{i,j} \; ; \; \boldsymbol{\pi} \mathbb{1}_J = \boldsymbol{\mu}, \boldsymbol{\pi}^\top \mathbb{1}_I = \boldsymbol{\nu} \right\}, \quad (\bar{\mathcal{P}}_\varepsilon)$$

$$= \max_{\mathbf{u} \in \mathbb{R}^I, \mathbf{v} \in \mathbb{R}^J} \sum_i \mathbf{u}_i \boldsymbol{\mu}_i + \sum_j \mathbf{v}_j \boldsymbol{\nu}_j - \varepsilon \sum_{i,j} \exp\left( \frac{\mathbf{u}_i + \mathbf{v}_j - \mathbf{c}_{i,j}}{\varepsilon} \right) \boldsymbol{\mu}_i \boldsymbol{\nu}_j, \quad \text{(for } \varepsilon > 0) \quad (\bar{\mathcal{D}}_\varepsilon)$$

$$= \max_{\mathbf{v} \in \mathbb{R}^J} \bar{H}_\varepsilon(\mathbf{v}) = \sum_{i \in I} \bar{h}_\varepsilon(x_i, \mathbf{v}) \boldsymbol{\mu}_i, \quad \text{where} \quad (\bar{\mathcal{S}}_\varepsilon)$$

$$\bar{h}_\varepsilon(x, \mathbf{v}) = \sum_{j \in J} \mathbf{v}_j \boldsymbol{\nu}_j + \begin{cases} -\varepsilon \log(\sum_{j \in J} \exp(\frac{\mathbf{v}_j - c(x, y_j)}{\varepsilon}) \boldsymbol{\nu}_j) - \varepsilon & \text{if } \varepsilon > 0, \\ \min_j (c(x, y_j) - \mathbf{v}_j) & \text{if } \varepsilon = 0, \end{cases} \quad (4)$$

The state-of-the-art method to solve the discrete regularized OT (i.e. when $\varepsilon > 0$) is Sinkhorn's algorithm [6, Alg.1], which has linear convergence rate [8]. It corresponds to a block coordinate maximization, successively optimizing $(\bar{\mathcal{D}}_\varepsilon)$ with respect to either $\mathbf{u}$ or $\mathbf{v}$. Each iteration of this algorithm is however costly, because it requires a matrix-vector multiplication. Indeed, this corresponds to a "batch" method where all the samples $(x_i)_i$ and $(y_j)_j$ are used at each iteration, which has thus complexity $O(N^2)$ where $N = \max(I, J)$. We now detail how to alleviate this issue using online stochastic optimization methods.

**Incremental Discrete Optimization when** $\varepsilon > 0$. Stochastic gradient descent (SGD), in which an index $k$ is drawn from distribution $\boldsymbol{\mu}$ at each iteration can be used to minimize the finite sum that appears in in $\bar{\mathcal{S}}_\varepsilon$. The gradient of that term $\bar{h}_\varepsilon(x_k, \cdot)$ can be used as a proxy for the full gradient in a standard gradient ascent step to maximize $\bar{H}_\varepsilon$.

When $\varepsilon > 0$, the finite sum appearing in $(\bar{\mathcal{S}}_\varepsilon)$ suggests to use incremental gradient methods—rather than purely stochastic ones—which are known to converge faster than SGD. We propose to use the stochastic averaged gradient (SAG) [19]. As SGD, SAG operates at each iteration by sampling a point $x_k$ from $\mu$, to compute the gradient corresponding to that sample for the current estimate $\mathbf{v}$. Unlike SGD, SAG keeps in memory a copy of that gradient. Another difference is that SAG applies a *fixed* length update, in the direction of the *average* of all gradients stored so far, which provides a better proxy of the gradient corresponding to the entire

---

**Algorithm 1** SAG for Discrete OT

**Input:** $C$
**Output:** $\mathbf{v}$

$\quad \mathbf{v} \leftarrow \mathbb{0}_J, \mathbf{d} \leftarrow \mathbb{0}_J, \forall i, \mathbf{g}_i \leftarrow \mathbb{0}_J$
$\quad \textbf{for } k = 1, 2, \ldots \textbf{ do}$
$\quad\quad \text{Sample } i \in \{1, 2, \ldots, I\} \text{ uniform.}$
$\quad\quad \mathbf{d} \leftarrow \mathbf{d} - \mathbf{g}_i$
$\quad\quad \mathbf{g}_i \leftarrow \boldsymbol{\mu}_i \nabla_v \bar{h}_\varepsilon(x_i, \mathbf{v})$
$\quad\quad \mathbf{d} \leftarrow \mathbf{d} + \mathbf{g}_i \; ; \; \mathbf{v} \leftarrow \mathbf{v} + C\mathbf{d}$
$\quad \textbf{end for}$

---

sum. This improves the convergence rate to $|\bar{H}_\varepsilon(\mathbf{v}_\varepsilon^\star) - \bar{H}_\varepsilon(\mathbf{v}_k)| = O(1/k)$, where $\mathbf{v}_\varepsilon^\star$ is a minimizer of $\bar{H}_\varepsilon$, at the expense of storing the gradient for each of the $I$ points. This expense can be mitigated by considering mini-batches instead of individual points. Note that the SAG algorithm is adaptive to strong-convexity and will be linearly convergent around the optimum. The pseudo-code for SAG is provided in Algorithm 1, and we defer more details on SGD for Section 4, in which it will be shown to play a crucial role. Note that the Lipschitz constant of all these terms is upperbounded by $L = \max_i \boldsymbol{\mu}_i / \varepsilon$.

**Numerical Illustrations on Bags of Word-Embeddings.** Comparing texts using a Wasserstein distance on their representations as clouds of word embeddings has been recently shown to yield state-of-the-art accuracy for text classification [11]. The authors of [11] have however highlighted that this accuracy comes at a large computational cost. We test our stochastic approach to discrete OT in this scenario, using the complete works of 35 authors (names in supplementary material). We use Glove word embeddings [14] to represent words, namely $\mathcal{X} = \mathcal{Y} = \mathbb{R}^{300}$. We discard all most frequent $1,000$ words that appear at the top of the file `glove.840B.300d` provided on the authors' website. We sample $N = 20,000$ words (found within the remaining huge dictionary of relatively rare words) from each authors' complete work. Each author is thus represented as a cloud of $20,000$ points in $\mathbb{R}^{300}$. The cost function $c$ between the word embeddings is the squared-Euclidean distance, re-scaled so that it has a unit empirical median on $2,000$ points sampled randomly among all vector embeddings. We set $\varepsilon$ to $0.01$ (other values are considered in the supplementary material). We compute all ($35 \times 34/2 = 595$) pairwise regularized Wasserstein distances using both the Sinkhorn algorithm and SAG. Following the recommendations in [19], SAG's stepsize is tested for 3 different settings, $1/L, 3/L$ and $5/L$. The convergence of each algorithm is measured by computing the $\ell_1$ norm of the gradient of the full sum (which also corresponds to the marginal violation of the primal transport solution that can be recovered with these dual variables[6]), as well as the $\ell_2$ norm of the

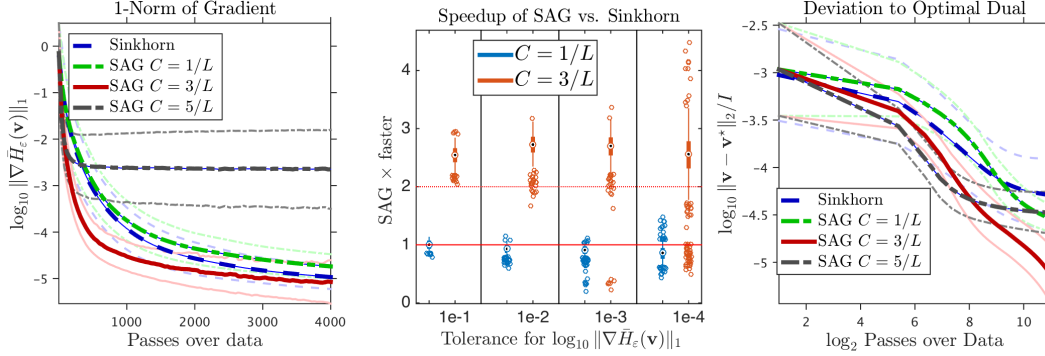

Figure 1: We compute all 595 pairwise word mover's distances [11] between 35 very large corpora of text, each represented as a cloud of $I = 20,000$ word embeddings. We compare the Sinkhorn algorithm with SAG, tuned with different stepsizes. Each pass corresponds to a $I \times I$ matrix-vector product. We used minibatches of size 200 for SAG. *Left plot*: convergence of the gradient $\ell_1$ norm (average and $\pm$ standard deviation error bars). A stepsize of $3/L$ achieves a substantial speed-up of $\approx 2.5$, as illustrated in the boxplots in the *center plot*. Convergence to $\mathbf{v}^\star$ (the best dual variable across all variables after $4,000$ passes) in $\ell_2$ norm is given in the *right plot*, up to $2,000 \approx 2^{11}$ steps.

deviation to the optimal scaling found after $4,000$ passes for any of the three methods. Results are presented in Fig. 1 and suggest that SAG can be more than twice faster than Sinkhorn on average for all tolerance thresholds. Note that SAG retains exactly the same parallel properties as Sinkhorn: all of these computations can be streamlined on GPUs. We used 4 Tesla K80 cards to compute both SAG and Sinkhorn results. For each computation, all $4,000$ passes take less than 3 minutes (far less are needed if the goal is only to approximate the Wasserstein distance itself, as proposed in [11]).

## 4 Semi-Discrete Optimal Transport

In this section, we assume that $\mu$ is an arbitrary measure (in particular, it needs not to be discrete) and that $\nu = \sum_{j=1}^{J} \boldsymbol{\nu}_j \delta_{y_j}$ is a discrete measure. This corresponds to the semi-discrete OT problem [1, 12]. The semi-dual problem $(\mathcal{S}_\varepsilon)$ is then a finite-dimensional maximization problem, written in expectation form as $W_\varepsilon(\mu, \nu) = \max_{\mathbf{v} \in \mathbb{R}^J} \mathbb{E}_X \left[ \bar{h}_\varepsilon(X, \mathbf{v}) \right]$ where $X \sim \mu$ and $\bar{h}_\varepsilon$ is defined in (4).

**Stochastic Semi-discrete Optimization.** Since the expectation is taken over an arbitrary measure, neither Sinkhorn algorithm nor incremental algorithms such as SAG can be used. An alternative is to approximate $\mu$ by an empirical measure $\hat{\mu}_N \overset{\text{def.}}{=} \frac{1}{N} \sum_{i=1}^{N} \delta_{x_i}$ where $(x_i)_{i=1,\ldots,N}$ are i.i.d samples from $\mu$, and computing $W_\varepsilon(\hat{\mu}_N, \nu)$ using the discrete methods (Sinkhorn or SAG) detailed in Section 3. However this introduces a discretization noise in the solution as the discrete problem is now different from the original one and thus has a different solution. Averaged SGD on the other hand does not require $\mu$ to be discrete and is thus perfectly adapted to this semi-discrete setting. The algorithm is detailed in Algorithm 2 (the expression for $\nabla \bar{h}_\varepsilon$ being given in Equation 4). The convergence rate is $O(1/\sqrt{k})$ thanks to averaging $\tilde{\mathbf{v}}_k$ [15].

---

**Algorithm 2** Averaged SGD for Semi-Discrete OT

**Input:** $C$
**Output:** $\mathbf{v}$
$\quad \tilde{\mathbf{v}} \leftarrow \mathbb{0}_J$, $\mathbf{v} \leftarrow \tilde{\mathbf{v}}$
$\quad$ **for** $k = 1, 2, \ldots$ **do**
$\quad\quad$ Sample $x_k$ from $\mu$
$\quad\quad \tilde{\mathbf{v}} \leftarrow \tilde{\mathbf{v}} + \frac{C}{\sqrt{k}} \nabla_v \bar{h}_\varepsilon(x_k, \tilde{\mathbf{v}})$
$\quad\quad \mathbf{v} \leftarrow \frac{1}{k} \tilde{\mathbf{v}} + \frac{k-1}{k} \mathbf{v}$
$\quad$ **end for**

---

**Numerical Illustrations.** Simulations are performed in $\mathcal{X} = \mathcal{Y} = \mathbb{R}^3$. Here $\mu$ is a Gaussian mixture (continuous density) and $\nu = \frac{1}{J} \sum_{j=1}^{J} \delta_{y_j}$ with $J = 10$ and $(x_j)_j$ are i.i.d. samples from another gaussian mixture. Each mixture is composed of three gaussians whose means are drawn randomly in $[0, 1]^3$, and their correlation matrices are constructed as $\Sigma = 0.01(R^T + R) + 3I_3$ where $R$ is $3 \times 3$ with random entries in $[0, 1]$. In the following, we denote $\mathbf{v}_\varepsilon^\star$ a solution of $(\mathcal{S}_\varepsilon)$, which is approximated by running SGD for $10^7$ iterations, 100 times more than those plotted, to ensure reliable convergence curves. Both plots are averaged over 50 runs, lighter lines show the variability in a single run.

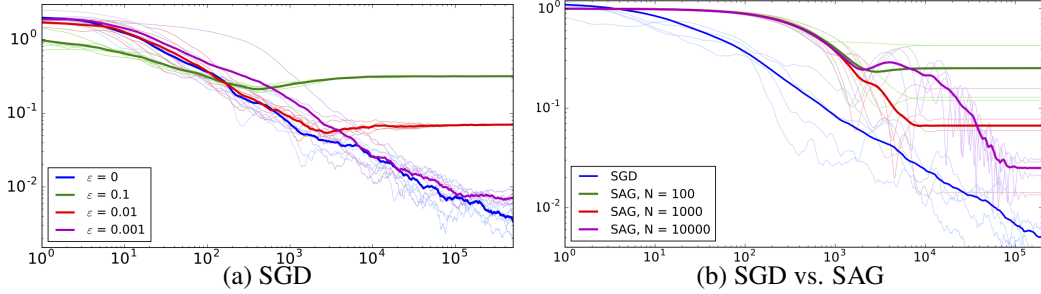

Figure 2: (a) Plot of $\|\mathbf{v}_k - \mathbf{v}_0^\star\|_2 / \|\mathbf{v}_0^\star\|_2$ as a function of $k$, for SGD and different values of $\varepsilon$ ($\varepsilon = 0$ being un-regularized). (b) Plot of $\|\mathbf{v}_k - \mathbf{v}_\varepsilon^\star\|_2 / \|\mathbf{v}_\varepsilon^\star\|_2$ as a function of $k$, for SGD and SAG with different number $N$ of samples, for regularized OT using $\varepsilon = 10^{-2}$.

Figure 2 (a) shows the evolution of $\|\mathbf{v}_k - \mathbf{v}_0^\star\|_2 / \|\mathbf{v}_0^\star\|_2$ as a function of $k$. It highlights the influence of the regularization parameters $\varepsilon$ on the iterates of SGD. While the regularized iterates converge faster, they do not converge to the correct unregularized solution. This figure also illustrates the convergence theorem of solution of $(\mathcal{S}_\varepsilon)$ toward those $(\mathcal{S}_0)$ when $\varepsilon \to 0$, which can be found in the supplementary material. Figure 2 (b) shows the evolution of $\|\mathbf{v}_k - \mathbf{v}_\varepsilon^\star\|_2 / \|\mathbf{v}_\varepsilon^\star\|_2$ as a function of $k$, for a fixed regularization parameter value $\varepsilon = 10^{-2}$. It compares SGD to SAG using different numbers $N$ of samples for the empirical measures $\hat{\mu}_N$. While SGD converges to the true solution of the semi-discrete problem, the solution computed by SAG is biased because of the approximation error which comes from the discretization of $\mu$. This error decreases when the sample size $N$ is increased, as the approximation of $\mu$ by $\hat{\mu}_N$ becomes more accurate.

## 5 Continuous optimal transport using RKHS

In the case where neither $\mu$ nor $\nu$ are discrete, problem $(\mathcal{S}_\varepsilon)$ is infinite-dimensional, so it cannot be solved directly using SGD. We propose in this section to solve the initial dual problem $(\mathcal{D}_\varepsilon)$, using expansions of the dual variables in two reproducing kernel Hilbert spaces (RKHS). Choosing dual variables (or test functions) in a RKHS is the fundamental assumption underlying the Maximum Mean Discrepancy (MMD)[22]. It is thus tempting to draw parallels between the approach in this section and the MMD. The two methods do not, however, share much beyond using RKHSs. Indeed, unlike the MMD, problem $(\mathcal{D}_\varepsilon)$ involves two different dual (test) functions $u$ and $v$, one for each measure; these are furthermore linked through a regularizer $\iota_{U_c}^\varepsilon$. Recall finally that contrarily to the semi-discrete setting, we can only solve the regularized problem here (i.e. $\varepsilon > 0$), since $(\mathcal{D}_\varepsilon)$ cannot be cast as an expectation maximization problem when $\varepsilon = 0$.

**Stochastic Continuous Optimization.** We consider two RKHS $\mathcal{H}$ and $\mathcal{G}$ defined on $\mathcal{X}$ and on $\mathcal{Y}$, with kernels $\kappa$ and $\ell$, associated with norms $\|\cdot\|_\mathcal{H}$ and $\|\cdot\|_\mathcal{G}$. Recall the two main properties of RKHS: (a) if $u \in \mathcal{H}$, then $u(x) = \langle u, \kappa(\cdot, x)\rangle_\mathcal{H}$ and (b) $\kappa(x, x') = \langle \kappa(\cdot, x), \kappa(\cdot, x')\rangle_\mathcal{H}$.

The dual problem $(\mathcal{D}_\varepsilon)$ is conveniently re-written in (3) as the maximization of the expectation of $f^\varepsilon(X, Y, u, v)$ with respect to the random variables $(X, Y) \sim \mu \otimes \nu$. The SGD algorithm applied to this problem reads, starting with $u_0 = 0$ and $v_0 = 0$,

$$(u_k, v_k) \stackrel{\text{def.}}{=} (u_{k-1}, v_{k-1}) + \frac{C}{\sqrt{k}} \nabla f_\varepsilon(x_k, y_k, u_{k-1}, v_{k-1}) \in \mathcal{H} \times \mathcal{G}, \qquad (5)$$

where $(x_k, y_k)$ are i.i.d. samples from $\mu \otimes \nu$. The following proposition shows that these $(u_k, v_k)$ iterates can be expressed as finite sums of kernel functions, with a simple recursion formula.

**Proposition 5.1.** *The iterates $(u_k, v_k)$ defined in (5) satisfy*

$$(u_k, v_k) \stackrel{\text{def.}}{=} \sum_{i=1}^{k} \alpha_i \big(\kappa(\cdot, x_i), \ell(\cdot, y_i)\big), \text{ where } \alpha_i \stackrel{\text{def.}}{=} \Pi_{B_r}\left(\frac{C}{\sqrt{i}}\left(1 - e^{\frac{u_{i-1}(x_i) + v_{i-1}(y_i) - c(x_i, y_i)}{\varepsilon}}\right)\right), \quad (6)$$

*where $(x_i, y_i)_{i=1\ldots k}$ are i.i.d samples from $\mu \otimes \nu$ and $\Pi_{B_r}$ is the projection on the centered ball of radius $r$. If the solutions of $(\mathcal{D}_\varepsilon)$ are in the $\mathcal{H} \times \mathcal{G}$ and if $r$ is large enough, the iterates $(u_k, v_k)$ converge to a solution of $(\mathcal{D}_\varepsilon)$.*

The proof of proposition 5.1 can be found in the supplementary material.

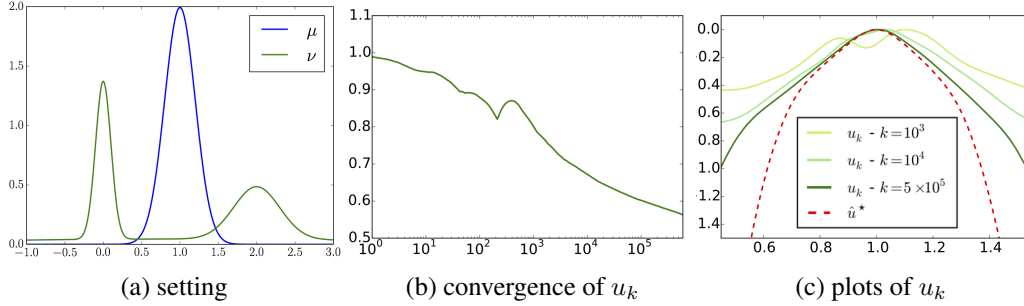

(a) setting        (b) convergence of $u_k$        (c) plots of $u_k$

Figure 3: (a) Plot of $\frac{d\mu}{dx}$ and $\frac{d\nu}{dx}$. (b) Plot of $\|\mathbf{u}_k - \hat{\mathbf{u}}^\star\|_2 / \|\hat{\mathbf{u}}^\star\|_2$ as a function of $k$ with SGD in the RKHS, for regularized OT using $\varepsilon = 10^{-1}$. (c) Plot of the iterates $u_k$ for $k = 10^3, 10^4, 10^5$ and the proxy for the true potential $\hat{u}^\star$, evaluated on a grid where $\mu$ has non negligible mass.

Algorithm 3 describes our kernel SGD approach, in which both potentials $u$ and $v$ are approximated by a linear combination of kernel functions. The main cost lies in the computation of the terms $u_{k-1}(x_k)$ and $v_{k-1}(y_k)$ which imply a quadratic complexity $O(k^2)$. Several methods exist to alleviate the running time complexity of kernel algorithms, e.g. random Fourier features [16] or incremental incomplete Cholesky decomposition [25].

---

**Algorithm 3** Kernel SGD for continuous OT

**Input:** $C$, kernels $\kappa$ and $\ell$
**Output:** $(\alpha_k, x_k, y_k)_{k=1,\dots}$
    **for** $k = 1, 2, \dots$ **do**
        Sample $x_k$ from $\mu$
        Sample $y_k$ from $\nu$
        $u_{k-1}(x_k) \stackrel{\text{def.}}{=} \sum_{i=1}^{k-1} \alpha_i \kappa(x_k, x_i)$
        $v_{k-1}(y_k) \stackrel{\text{def.}}{=} \sum_{i=1}^{k-1} \alpha_i \ell(y_k, y_i)$
        $\alpha_k \stackrel{\text{def.}}{=} \frac{C}{\sqrt{k}} \left(1 - e^{\frac{u_{k-1}(x_k) + v_{k-1}(y_k) - c(x_k, y_k)}{\varepsilon}}\right)$
    **end for**

---

Kernels that are associated with dense RHKS are called universal [23] and can approach any arbitrary potential. In Euclidean spaces $\mathcal{X} = \mathcal{Y} = \mathbb{R}^d$, where $d > 0$, a natural choice of universal kernel is the kernel defined by $\kappa(x, x') = \exp(-\|x - x'\|^2/\sigma^2)$. Tuning its bandwidth $\sigma$ is crucial to obtain a good convergence of the algorithm.

Finally, let us note that, while entropy regularization of the primal problem $(\mathcal{P}_\varepsilon)$ was instrumental to be able to apply semi-discrete methods in Sections 3 and 4, this is not the case here. Indeed, since the kernel SGD algorithm is applied to the dual $(\mathcal{D}_\varepsilon)$, it is possible to replace $\mathrm{KL}(\pi|\mu \otimes \nu)$ appearing in $(\mathcal{P}_\varepsilon)$ by other regularizing divergences. A typical example would be a $\chi^2$ divergence $\int_{\mathcal{X} \times \mathcal{Y}} (\frac{d\pi}{d\mu d\nu}(x, y))^2 d\mu(x) d\nu(y)$ (with positivity constraints on $\pi$).

**Numerical Illustrations.** We consider optimal transport in 1D between a Gaussian $\mu$ and a Gaussian mixture $\nu$ whose densities are represented in Figure 3 (a). Since there is no existing benchmark for continuous transport, we use the solution of the semi-discrete problem $W_\varepsilon(\mu, \hat{\nu}_N)$ with $N = 10^3$ computed with SGD as a proxy for the solution and we denote it by $\hat{u}^\star$. We focus on the convergence of the potential $u$, as it is continuous in both problems contrarily to $v$. Figure 3 (b) represents the plot of $\|\mathbf{u}_k - \hat{\mathbf{u}}^\star\|_2/\|\hat{\mathbf{u}}^\star\|_2$ where $\mathbf{u}$ is the evaluation of $u$ on a sample $(x_i)_{i=1\dots N'}$ drawn from $\mu$. This gives more emphasis to the norm on points where $\mu$ has more mass. The convergence is rather slow but still noticeable. The iterates $u_k$ are plotted on a grid for different values of $k$ in Figure 3 (c), to emphasize the convergence to the proxy $\hat{u}^\star$. We can see that the iterates computed with the RKHS converge faster where $\mu$ has more mass, which is actually where the value of $u$ has the greatest impact in $F_\varepsilon$ ($u$ being integrated against $\mu$).

## Conclusion

We have shown in this work that the computations behind (regularized) optimal transport can be considerably alleviated, or simply enabled, using a stochastic optimization approach. In the discrete case, we have shown that incremental gradient methods can surpass the Sinkhorn algorithm in terms of efficiency, taken for granted that the (constant) stepsize has been correctly selected, which should be possible in practical applications. We have also proposed the first known methods that can address the challenging semi-discrete and continuous cases. All of these three settings can open new perspectives for the application of OT to high-dimensional problems.

**Acknowledgements** GP was supported by the European Research Council (ERC SIGMA-Vision); AG by Région Ile-de-France; MC by JSPS grant 26700002.

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
