[Supplementary Material · StochasticOT-NIPS-supplementary.pdf]

# Stochastic Optimization for Large-scale Optimal Transport
## *Supplementary Material*

**Aude Genevay**
CEREMADE, Université Paris Dauphine
INRIA – Mokaplan project-team
`genevay@ceremade.dauphine.fr`

**Marco Cuturi**
CREST, ENSAE
Université Paris-Saclay
`marco.cuturi@ensae.fr`

**Gabriel Peyré**
CNRS and DMA, École Normale Supérieure
INRIA – Mokaplan project-team
`gabriel.peyre@ens.fr`

**Francis Bach**
INRIA – Sierra project-team
DI, École Normale Supérieure
`francis.bach@inria.fr`

Let us recall the primal, dual and semi-dual problems

$$W_\varepsilon(\mu,\nu) \overset{\text{def.}}{=} \min_{\pi\in\Pi(\mu,\nu)} \int_{\mathcal{X}\times\mathcal{Y}} c(x,y)\mathrm{d}\pi(x,y) + \varepsilon\,\mathrm{KL}(\pi|\mu\otimes\nu), \qquad (\mathcal{P}_\varepsilon)$$

$$= \max_{u\in\mathcal{C}(\mathcal{X}),v\in\mathcal{C}(\mathcal{Y})} F_\varepsilon(u,v) \overset{\text{def.}}{=} \int_{\mathcal{X}} u(x)\mathrm{d}\mu(x) + \int_{\mathcal{Y}} v(y)\mathrm{d}\nu(y) - \iota^\varepsilon_{U_c}(u,v), \qquad (\mathcal{D}_\varepsilon)$$

$$= \max_{v\in\mathcal{C}(\mathcal{Y})} H_\varepsilon(v) \overset{\text{def.}}{=} \int_{\mathcal{X}} v^{c,\varepsilon}(x)\mathrm{d}\mu(x) + \int_{\mathcal{Y}} v(y)\mathrm{d}\nu(y) - \varepsilon. \qquad (\mathcal{S}_\varepsilon)$$

## 1 Convergence of $(\mathcal{S}_\varepsilon)$ as $\varepsilon \to 0$

The convergence of the solution of $(\mathcal{P}_\varepsilon)$ toward a solution of $(\mathcal{P}_0)$ as $\varepsilon \to 0$ is proved in [1]. The convergence of solutions of $(\mathcal{D}_\varepsilon)$ toward solutions of $(\mathcal{D}_0)$ as $\varepsilon \to 0$ is proved for the special case of discrete measures in [2]. To the best of our knowledge, the behavior of $(\mathcal{S}_\varepsilon)$ has not been studied in the literature, and we propose a convergence result in the case where $\nu$ is discrete, which is the setting in which this formulation is most advantageous.

**Proposition 1.1.** *We assume that $\forall y \in \mathcal{Y}$, $c(\cdot,y) \in L^1(\mu)$, that $\nu = \sum_{j=1}^{J} \boldsymbol{\nu}_j \delta_{y_j}$, and we fix $x_0 \in \mathcal{X}$,. For all $\varepsilon > 0$, let $v^\varepsilon$ be the unique solution of $(\mathcal{S}_\varepsilon)$ such that $v^\varepsilon(x_0) = 0$. Then $(v^\varepsilon)_\varepsilon$ is bounded and all its converging sub-sequences for $\varepsilon \to 0$ are solutions of $(\mathcal{S}_0)$.*

We first prove a useful lemma.

**Lemma 1.2.** *If $\forall y$, $x \mapsto c(x,y) \in L^1(\mu)$ then $H_\varepsilon$ converges pointwise to $H_0$.*

*Proof.* Let $\alpha_j(x) \overset{\text{def.}}{=} v_j - c(x,y_j)$ and $j^\star \overset{\text{def.}}{=} \arg\max_j \alpha_j(x)$.
On the one hand, since $\forall j$, $\alpha_j(x) \leq \alpha_{j^\star}(x)$ we get

$$\varepsilon\log(\sum_{j=1}^{J} e^{\frac{\alpha_j(x)}{\varepsilon}}\nu_j) = \varepsilon\log(e^{\frac{\alpha_{j^\star}(x)}{\varepsilon}}\sum_{j=1}^{J} e^{\frac{\alpha_j(x)-\alpha_{j^\star}(x)}{\varepsilon}}\nu_j) \leq \alpha_{j^\star}(x) + \varepsilon\log(\sum_{j=1}^{J}\nu_j) = \alpha_{j^\star}(x) \quad (1)$$

On the other hand, since $\log$ is increasing and all terms in the sum are non negative we have

$$\varepsilon\log(\sum_{j=1}^{J} e^{\frac{\alpha_j(x)}{\varepsilon}}\nu_j) \geq \varepsilon\log(e^{\frac{\alpha_{j^\star}(x)}{\varepsilon}}\nu_{j^\star}) = \alpha_{j^\star}(x) + \varepsilon\log(\nu_{j^\star}) \overset{\varepsilon\to 0}{\longrightarrow} \alpha_{j^\star}(x) \qquad (2)$$

Hence $\varepsilon \log(\sum_{j=1}^{J} e^{\frac{\alpha_j(x)}{\varepsilon}} \nu_j) \xrightarrow{\varepsilon \to 0} \alpha_{j^\star}(x)$ and $\varepsilon \log(\sum_{j=1}^{J} e^{\frac{\alpha_j(x)}{\varepsilon}} \nu_j) \leq \alpha_{j^\star}(x)$.

Since we assumed $x \mapsto c(x, y_j) \in L^1(\mu)$, then $\alpha_{j^\star} \in L^1(\mu)$ and by dominated convergence we get that $H_\varepsilon(v) \xrightarrow{\varepsilon \to 0} H_0(v)$. $\qquad\square$

*Proof of Proposition 1.1.* First, let's prove that $(v_\varepsilon)_\varepsilon$ has a converging subsequence. With similar computations as in Proposition 2.1 we get that $v_\varepsilon(y_i) = -\varepsilon \log(\int_{\mathcal{X}} e^{\frac{u_\varepsilon(x) - c(x, y_i)}{\varepsilon}} d\mu(x))$. We denote by $\tilde{v}_\varepsilon$ the c-transform of $u_\varepsilon$ such that $\tilde{v}_\varepsilon(y_i) = \min_{x \in \mathcal{X}} c(x, y_i) - u_\varepsilon(x)$. From standard results on optimal transport (see [4], p.11) we know that $| \tilde{v}_\varepsilon(y_i) - \tilde{v}_\varepsilon(y_j) | \leq \omega(\|y_i - y_j\|)$ where $\omega$ is the modulus of continuity of the cost $c$. Besides, using once again the soft-max argument we can bound $| v_\varepsilon(y) - \tilde{v}_\varepsilon(y) |$ by some constant C.

Thus we get that :

$$
\begin{aligned}
| v_\varepsilon(y_i) - v_\varepsilon(y_j) | &\leq | v_\varepsilon(y_i) - \tilde{v}_\varepsilon(y_i) | + | \tilde{v}_\varepsilon(y_i) - \tilde{v}_\varepsilon(y_j) | + | \tilde{v}_\varepsilon(y_j) - v_\varepsilon(y_j) | \quad (3) \\
&\leq C + \omega(\|y_i - y_j\|) + C \quad (4)
\end{aligned}
$$

Besides, the regularized potentials are unique up to an additive constant. Hence we can set without loss of generality $v_\varepsilon(y_0) = 0$. So from the previous inequality yields :

$$v_\varepsilon(y_i) \leq 2C + \omega(\|y_i - y_0\|) \quad (5)$$

So $v_\varepsilon$ is bounded on $\mathbb{R}^J$ which is a compact set and thus we can extract a subsequence which converges to a certain limit that we denote by $\bar{v}$.

Let $v^\star \in \arg\max_v H_0$. To prove that $\bar{v}$ is optimal, it suffices to prove that $H_0(v^\star) \leq H_0(\bar{v})$. By optimality of $v_\varepsilon$,

$$H_\varepsilon(v^\star) \leq H_\varepsilon(v_\varepsilon) \quad (6)$$

The term on the left-hand side of the inequality converges to $H_0(v^\star)$ since $H_\varepsilon$ converges pointwise to $H_0$. We still need to prove that the right-hand term converges to $H_0(\bar{v})$.

By the Mean Value Theorem, there exists $\tilde{v}_\varepsilon \overset{\text{def.}}{=} (1 - t^\varepsilon)v_\varepsilon + t^\varepsilon \bar{v}$ for some $t^\varepsilon \in [0, 1]$ such that

$$| H_\varepsilon(v_\varepsilon) - H_\varepsilon(\bar{v}) | \leq \|\nabla H_\varepsilon(\tilde{v}_\varepsilon)\| \, \|v_\varepsilon - \bar{v}\| \quad (7)$$

The gradient of $H_\varepsilon$ reads

$$\nabla_v H_\varepsilon(v) = \nu - \pi(v) \quad (8)$$

where $\pi_i(v) = \dfrac{\int_{\mathcal{X}} e^{\frac{v_i - c(x, y_i)}{\varepsilon}} \nu_i d\mu(x)}{\int_{\mathcal{X}} \sum_{j=1}^{J} e^{\frac{v_j - c(x, y_j)}{\varepsilon}} \nu_j d\mu(x)}$

It is the difference of two elements in the simplex thus it is bounded by a constant $C$ independently of $\varepsilon$.

Using this bound in (7) get

$$H_\varepsilon(\bar{v}) - C \|v_\varepsilon - \bar{v}\| \leq H_\varepsilon(v_\varepsilon) \leq H_\varepsilon(\bar{v}) + C \|v_\varepsilon - \bar{v}\| \quad (9)$$

By pointwise convergence of $H_\varepsilon$ we know that $H_\varepsilon(\bar{v}) \to H_0(\bar{v})$, and since $\bar{v}$ is a limit point of $v_\varepsilon$ we can conclude that the left and right hand term of the inequality converge to $H_0(\bar{v})$. Thus we get $H_\varepsilon(v_\varepsilon) \to H_0(\bar{v})$. $\qquad\square$

## 2  Discrete-Discrete Setting

The list of authors we consider is: KEATS, CERVANTES, SHELLEY, WOOLF, NIETZSCHE, PLUTARCH, FRANKLIN, COLERIDGE, MAUPASSANT, NAPOLEON, AUSTEN, BIBLE, LINCOLN, PAINE, DELAFONTAINE, DANTE, VOLTAIRE, MOORE, HUME, BURROUGHS, JEFFERSON, DICKENS, KANT, ARISTOTLE, DOYLE, HAWTHORNE, PLATO, STEVENSON, TWAIN, IRVING, EMERSON, POE, WILDE, MILTON, SHAKESPEARE.

Figure 1: Comparisons between the Sinkhorn algorithm and SAG, tuned with different stepsizes, using different regularization strengths. The setting is identical to that used in Figure 1. Note that to prevent numerical overflow when using very small regularizations, the metric is thresholded such that rescaled costs $c(x, y_j)/\varepsilon$ are not bigger than $\log(10^{200})$.

## 3 Proof of convergence of stochastic gradient descent in the RKHS

The SGD algorithm applied to the regularized continuous OT problem reads, starting with $u_0 = 0$ and $v_0 = 0$,

$$(u_k, v_k) \stackrel{\text{def.}}{=} (u_{k-1}, v_{k-1}) + \frac{C}{\sqrt{k}} \nabla f_\varepsilon(x_k, y_k, u_{k-1}, v_{k-1}) \in \mathcal{H} \times \mathcal{G}, \tag{10}$$

where $(x_k, y_k)$ are i.i.d. samples from $\mu \otimes \nu$. The following proposition shows that these $(u_k, v_k)$ iterates can be expressed as finite sums of kernel functions, with a simple recursion formula.

**Proposition 3.1.** *The iterates $(u_k, v_k)$ defined in* (10) *satisfy*

$$(u_k, v_k) \stackrel{\text{def.}}{=} \sum_{i=1}^{k} \alpha_i(\kappa(\cdot, x_i), \ell(\cdot, y_i)), \text{ where } \alpha_i \stackrel{\text{def.}}{=} \Pi_{B_r}\left( \frac{C}{\sqrt{i}} \left( 1 - e^{\frac{u_{i-1}(x_i) + v_{i-1}(y_i) - c(x_i, y_i)}{\varepsilon}} \right) \right), \tag{11}$$

*where $(x_i, y_i)_{i=1 \ldots k}$ are i.i.d samples from $\mu \otimes \nu$ and $\Pi_{B_r}$ is the projection on the centered ball of radius $r$. If the solutions of $(\mathcal{D}_\varepsilon)$ are in the $\mathcal{H} \times \mathcal{G}$ and if $r$ is large enough, the iterates $(u_k, v_k)$ converge to a solution of $(\mathcal{D}_\varepsilon)$.*

*Proof.* Rewriting $u(x)$ and $v(y)$ as scalar products in $f^\varepsilon(X, Y, u, v)$ yields

$$f^\varepsilon(x, y, u, v) = \langle u, \kappa(x, \cdot) \rangle_\mathcal{H} + \langle v, \ell(y, \cdot) \rangle_\mathcal{G} - \varepsilon \exp\left( \frac{\langle u, \kappa(x, \cdot) \rangle_\mathcal{H} + \langle v, \ell(y, \cdot) \rangle_\mathcal{G} - c(x, y)}{\varepsilon} \right).$$

Plugging this formula in iteration (10) yields : $(u_k, v_k) = u_{k-1} + \alpha_k(\kappa(\cdot, x_k), \ell(\cdot, y_k))$ ,where $\alpha_k$ is defined in (11). Since the parameters $(\alpha_i)_{i<k}$ are not updated at iteration $k$, we get the announced formula. Putting a bound on the iterates of $\alpha$ by projecting on $B_r$ ensures convergence [3]. $\qquad \square$