[Reviews · NeurIPS 2016]

Reviewer 1

Summary

Optimal transport (OT) is a way to compare prob. distributions, and is computational expensive operation. The authors have proposed a stochastic optimization by only requiring that they are able to draw from the distributions. This way they do not need to discretise the densities and instead use the fact that the dual OT can be recast as max. of expectation and entropic regularization of the primal OT can be solved via faster algorithm. This way the authors can compare discrete-discrete, discrete-continuous and by using SGD over a reproducing kernel Hilbert Space they can solve the continuous-continuous problem as well!

Qualitative Assessment

The authors have proposed a method which does not require discretize of densities, by finding two key insights: recasting the dual OT problem and entropic regularization of the primal OT problem. This way they are able to get, when comparing discrete distributions, a method which is faster than the current state of the art. Their method also gets better performance when comparing discrete-continuous distributions, and finally give the only known method of optimizing continuous-continuous distributions, via SGD over a reproducing kernel Hilbert space. This is an amazing break-through and will prove to be very useful.

Confidence in this Review

2-Confident (read it all; understood it all reasonably well)


Reviewer 2

Summary

The authors study the problem of learning the optimal transport between two measures. The potential applications of the paper are concerned by 'large-scale' problems, leading to a natural use of a stochastic gradient strategy. The contribution can be,roughly speaking , splitted into three parts: 1) the dual formulation of the OT problem is traduced in an optimization problem with an expectation $(\mathcal{D}_{\epsilon})$ (See Proposition 2.1). From a lack of strong convexity of the objective function, authors naturally introduce a Kullback-Leibler entropy penalization (with a coefficient $\epsilon$ that balances the effect of this penalization). 2) the dual (penalized) problem is solved with a stochastic gradient procedure. Several methods are possible: SGD, Polyak averaging, Av S.G.D. The authors choose to use Av. S.G.D. 3) Simulations are somewhat convincing, but I am not a specialist of Bag of words embbedings problem. Therefore, maybe I should not be trusted in on this point ;) Authors propose several frameworks: discrete, semi-discrete, continuous OT problems. The last ones are dealt with RKHS.

Qualitative Assessment

My evaluation of the paper is balanced. Pros: The writing of the paper is very clear. The algorithms are very natural and suspected to work well on large scale OT problems. The dual formulation is a good idea to handle a stochastic algorithm. Everything is ok with all that points. But... Cons: 1/ The mirror descent strategy is not new. Some references should be added somewhere in the paper. Even though not already applied to O.T. problems, it has been studied by several authors (not me! ;) ) when Bregman divergences are involved in the optimization problem. I guess it has also been studied with stochastic algorithms. 2/ There is no real theoretical analysis in the paper. The influence of the size of epsilon is certainly very important for convergence rates of the (Av.) S.G.D. described here because it quantifies the size of the strong convexity. A natural extension should use a cooling strategy eps-> 0 when n is increasing, to jointly associate the effect of the a.s. convergence of the S.G.D with the concentration of (S_epsilon) (briefly described in the Appendix). Of course, such a study could represent a good contribution for a top level statistical journal, and is not attended for NIPS. Nevertheless, the authors could include in their paper some theoretical insights (with a fixed epsilon for example). Now, I provide a linear description of my comments: L.16: instead of Sinkhorn's method, we can also use other optimization procedures of Teboulle to solve O.T. Sinkhorn's methods are the state of the art for this kind of problems only because the authors concerned by this problem have not read "Interior gradient and proximal methods for convex optimization" of Auslender and Teboulle. This kind of method could certainly be applied in this framework. L.80: please discuss on the lack of strong convexity here L.99: read the sentence ;) Section 2: somewhere should be mentionned that Sinkhorn's method has few theoretical guarantees (rates) L.129: If I remember well, I have seen somewhere that it is possible to assess some quantitative results stronger than the one of the Appendix, in terms of epsilon and rates. ( e^{-\lambda \epsilon^{-1}} ?) in Cominetti and San Martin, 92? L. 161->186: The choice of the stepsize is the icing on the cake of all stochastic gradient algorithms. As far as I can read, the authors have not written clearly somewhere what is this step size for them. It is only hidden in the description of Algorithm 1. The constant step size choice should be discussed somewhere, a stopping criterion should be described. In Algorithm 1, why i is choosen uniformly among \{1,\ldots I\}? Is it possible to use $\mu_i$ to sample i at each step k, and then update the distribution \mu? L. 192: I do not know what is Glove word embeddings. Some insights on the way $c$ is built should be helpful. L. 198: epsilon is chosen very small, leading to a very weak strong convexity of the OT problem. Some comments are welcome. Algorithm 2: this time, the step size used is k^{-1/2}. Again, some comments on this choice should be performed somewhere, and theoretical insights would be helpful for the reader.

Confidence in this Review

3-Expert (read the paper in detail, know the area, quite certain of my opinion)


Reviewer 3

Summary

This paper presents methods for the stochastic optimization of large-scale (regularized) optimal transport (OT), which is built upon the dual smoothed representation of entropic regularized OT [Cuturi and Peyré, 2016] (not cited). The paper shows that the dual smoothed objective (appeared in [Cuturi and Peyré, 2016]) can be rewritten differently as a finite separable sum, thus immediately admits stochastic methods by sampling one separate term at a time. As such, the paper explores three different settings of OT, where one or both measures are discrete or continuous, each with one or some illustrative numerical examples. In particular, it proposes to use SAG [16] for the discrete-discrete setting, SGD for the discrete-continuous setting, and RKHS parametrization for the continuous-continuous setting. In this paper, the convergence complexities are stated inaccurate (see my explanation below), experiments are done very briefly, some important claims are not well justified. The three research problems attempted certainly go beyond an 8-page paper could embrace. Cuturi, Marco, and Gabriel Peyré. "A smoothed dual approach for variational Wasserstein problems." SIAM Journal on Imaging Sciences 9.1 (2016): 320-343.

Qualitative Assessment

(Update!) The authors have responded to most of my questions in the rebuttal. Given I was the only one who gave a negative score, I would like to summarize my take on their responses. 1. They indeed resolves most of my concerns in the rebuttal, which could have been appeared in their original submission. I believe as long as my questions got cleared out, the paper will meet the threshold of technical quality that NIPS asks for. Particularly, I would like to see: (a) Clarify that semi-dual formulation is not strongly convex. This message is not in the original paper and can be a plausible excuse that they don't specify the iteration complexity. (b) Show CPU time comparison between SAG and Sinkhorn per iteration. (c) Discuss the relation between their RKHS OT approach and the famous MMD kernel approach. (d) Rephrase their contribution section to avoid inaccurate statements. (e) Add sensitivity study on ɛ Given the authors have carefully answered most of my questions, I will raise the technical quality score to 3. Other scores are unchanged. 2. Some views that I made in the review still remains unchanged. But I will let others to decide whether they are valid reasons to reject this paper. (a) The originality is limited because the semi-dual formulation (new) is only one-step further than the smoothed dual formulation (already existed, but not clearly stated in the paper). (b) For the D-D case, given there is no theoretical justification that SAG is better than Sinkhorn. The empirical justification is quite brief, not so convincing at this point. (c) My view on "The authors study three problems in a single 8-page paper. Although they share the same optimal transport formulation, the actual approaches do not really hold any tight connections." ***** As I summarized, this paper has many issues. Four major problems are (1) The authors study three problems in a single 8-page paper. Although they share the same optimal transport formulation, the actual approaches do not really hold any tight connections. As a result, this paper leaves many questions unanswered for their approach on each problem, which I’d like to list them one by one later. Some questions I listed are very important, and should be answered to justify a valid algorithmic contribution beyond a half-baked idea. (2) The formulation (Prop 2.1) are not completely new, Eq. (D_\varepsilon) and later Eq. (\bar{D}_\varepsilon) already exist in [Cuturi and Peyré, 2016]. Thus, the originality is quite limited. (3) Some claimed contributions of this paper are not properly justified or ambiguous. (4) Experiments are not very useful for understanding the convergence properties of each algorithm. As such, this paper could be improved by focusing on one of the three settings, clarifying and justifying its novel contributions, and empirically demonstrating its superiority or validity more carefully. More detailed comments are as follows. (a) Discrete-discrete setting. This is the most practical setting I want to focus in this review. First of all, there is no evidence in the paper that the SAG approach gives a convergent solution for classical optimal transport. No experiments has done in this regard. Therefore, claims in Line 73 “These methods can be used to solve both classical OT problems and their entropic-regularized versions (which enjoy faster convergence properties). ” are not properly justified. Second, I want to clarify the differences between SAG and SGD. SAG and many other variance reducing variants are quite different from SGD in terms of their iteration complexity. SAG is linearly convergent (or exponentially convergent ) for smooth, strongly convex problems, whose exponential rate depends on the data size N. SGD has much worse convergence rate, but the iteration complexity does NOT depend on the data size. As such, SAG is more closer to full gradient approach, rather than SGD (lucidly explained in [16]). SAG, like Sinkhorn iterations, enjoys linear convergence for regularized optimal transport [4,6] due to the use of entropic regularization. In order to justify one is faster than another, one has to either show theoretically and empirically one has a better exponential rate than the other (Please see what has been done for evaluating SAG [16, Table 1], where comparison between SAG and full gradient are made.) Because the iteration complexity of SAG depends on the problem’s conditional number (strongly convex constant and Lipschitz constant), which actually depends on the regularizing parameter ɛ. The comparison of complexities becomes even subtle. Is it possible that SAG becomes slower than Sinkhorn iteration for some ɛ ? Therefore, the statements in Line 78 “while still enjoying a convergence rate O(1/k)” is inaccurate. On one hand, the paper implies SAG has very cheap computation complexity per iteration, which is true, on the other hand, it does not specify the iteration complexity accurately (the dependencies upon N, L and ɛ). In fact, SAG is more expensive in terms of data passes than Sinkhorn. Third, the paper conducts an experiment comparing SAG and Sinkhorn iterations. Three flaws are (1) as mentioned earlier, sensitivity on ɛ is not explored. ɛ is a crucial parameter for making entropic regularization successful. (2) empirical exponential rates are not compared. (3) comparisons are made against passes over data, rather than the actual CPU/GPU time. The iteration of the SAG approach requires the computation of functions exp() and log(), which could be far more expensive than the Sinkhorn iteration. Therefore, the claims in Line 204 “SAG can be more than twice faster than Sinkhorn on average …” is not proper. Fourth, as mentioned in [Benamou, 2015] section 1.3, Sinkhorn iteration has a maximal possible iteration number for a small ɛ due to some quantities manipulated in Sinkhorn become smaller than machine precision. Does the SAG approach have this similar issue? Benamou, Jean-David, et al. "Iterative bregman projections for regularized transportation problems." SIAM Journal on Scientific Computing 37.2 (2015): A1111-A1138. (b) Discrete-Continuous Setting The paper claims in Line 80 “the SGD approach is numerically advantageous over the brute force approach consisting in sampling first …”. But their empirical validation is rather weak. Because (1) the experimental setting in which a huge number of iterations are used, is advantageous to SGD which converges rather slow. More iterations are clearly not useful for a problem with intrinsic discretization error, but one can increase N to reduce error just like the way SGD asks for more iterations. (2) The ground-truth v^\star “is approximated numerically by running SGD with a large enough number of iterations”. How large is large enough? Please specify the actual number. The logarithmic errors plotted in Figure 2 are known quite sensitive to the small changes of v^\star. As such, it makes the comparison even unfair. (3) Because entropic regularized OT is already an approximation to classical OT, I didn’t see much value to have a very accurate approximation to another approximation. (c) Continuous-Continuous Setting I didn’t see any practical value or motivation in this approach. It does not compute the exact solution for regularized OT, and converges slowly. In fact, if u and v are chosen from RKHS rather than some family of continuous functions, the dual OT problem is to compute maximum mean discrepancy, an extensively studied problem in the community of kernel methods. With RKHS embeddings, what’s the motivation to further consider the entropic regularization in regularizing the objective of maximum mean discrepancy? Please see for example [Gretton, 2012]. Gretton, Arthur, et al. "A kernel two-sample test." The Journal of Machine Learning Research 13.1 (2012): 723-773. I try to give some positive words in the end. I think the SAG approach for Discrete-Discrete setting is an important direction and should be empirically investigated more if there is truly any practical value found in it. The complexity subtleties should be clarified and made easy and accessible to reader, so one can appreciate this idea to some degree. Simply put, for a regularized OT setup (N, L, ɛ), what is the iteration complexity? What is the computation time per iteration? How they compare to Sinkhorn? Given an OT with m,n support sizes, the current approach mainly considers only one of n,m is large. What if both m and n are large? Is it possible to use doubly stochastic primal-dual approaches, e.g. [Yu, 2015] [Zhang, 2014]? Stochastic optimization of OT is an important direction, and I encourage the authors to improve their work, and re-submit with higher solidility in the future. Yu, Adams Wei, Qihang Lin, and Tianbao Yang. "Doubly Stochastic Primal-Dual Coordinate Method for Regularized Empirical Risk Minimization with Factorized Data." arXiv preprint arXiv:1508.03390 (2015). Zhang, Yuchen, and Lin Xiao. "Stochastic primal-dual coordinate method for regularized empirical risk minimization." arXiv preprint arXiv:1409.3257(2014).

Confidence in this Review

3-Expert (read the paper in detail, know the area, quite certain of my opinion)


Reviewer 4

Summary

This paper propose to use stochastic optimization to solve optimal transport problems. The authors study three types of optimal transport problems: discrete, semi-discrete, and continuous, with the assumption that both distributions can be sampled. In the discrete case, they apply SAG, a variance reduced gradient method for minimizing finite-sum, to the semi-dual problem. In the semi-discrete case, they apply SGD with averaging to the semi-dual problem. In the continuous case, they utilize kernel SGD to optimize the dual problem. The correctness/effectiveness of algorithms are justified by analysis and experiments.

Qualitative Assessment

Overall, I enjoy reading this paper and find it reaches the quality standard for NIPS. Pros: -- The key idea is forceful and well supported. The three cases provide a complete and persuasive story of utilizing stochastic optimization for optimal transport. -- There is sufficient amount of novelty. While the idea of SAG or SGD is not new, their application in optimal transport setting is. Moreover, the SGD idea leads to solutions of semi-discrete and continuous OT problems without discretization. -- The figures and algorithm tables are very clear and nice-looking. Cons (or questions to be addressed): -- For the semi-discrete or continuous OT cases, methods using discretization (if there are any) should be compared against. The authors warn about their discretization errors, but do not show their performance. Since these methods are the only baseline, it is important to know if the proposed algorithms have any performance gain besides being conceptually attractive. -- Could you comment on why in the kernel SGD experiment the algorithm converges so slowly? As this is only a toy problem, I am worried that this algorithm is impractical in practice. Minor comment: in Algorithm 3, should \alpha_n be replaced by \alpha_k?

Confidence in this Review

2-Confident (read it all; understood it all reasonably well)